# Triangular Gross-Pitaevskii breathers and Damski-Chandrasekhar shock waves

M. Olshanii[1*], D. Deshommes[1], J. Torrents[2], M. Gonchenko[3], V. Dunjko[1], G. E. Astrakharchik[4]

**1** Department of Physics, University of Massachusetts Boston, Boston Massachusetts 02125, USA
**2** Departament de Física de la Matèria Condensada, Universitat de Barcelona, Martí i Franquès 1, 08028 Barcelona, Spain
**3** Departament de Matemàtiques, Universitat Politècnica de Catalunya, E08028 Barcelona, Spain
**4** Departament de Física, Universitat Politècnica de Catalunya, E08034 Barcelona, Spain *
maxim.olchanyi@umb.edu

May 6, 2021

## Abstract

The recently proposed map [arXiv:2011.01415] between the hydrodynamic equations governing the two-dimensional triangular cold-bosonic breathers [Phys. Rev. X 9, 021035 (2019)] and the high-density zero-temperature triangular free-fermionic clouds, both trapped harmonically, perfectly explains the former phenomenon but leaves uninterpreted the nature of the initial ($t = 0$) singularity. This singularity is a density discontinuity that leads, in the bosonic case, to an infinite force at the cloud edge. The map itself becomes invalid at times $t < 0$. A similar singularity appears at $t = T/4$, where $T$ is the period of the harmonic trap, with the Fermi-Bose map becoming invalid at $t > T/4$. Here, we first map—using the scale invariance of the problem—the trapped motion to an untrapped one. Then we show that in the new representation, the solution [arXiv:2011.01415] becomes, along a ray in the direction normal to one of the three edges of the initial cloud, a freely propagating one-dimensional shock wave of a class proposed by Damski in [Phys. Rev. A 69, 043610 (2004)]. There, for a broad class of initial conditions, the one-dimensional hydrodynamic equations can be mapped to the inviscid Burgers' equation, which is equivalent to a nonlinear transport equation. More specifically, under the Damski map, the $t = 0$ singularity of the original problem becomes, *verbatim*, the initial condition for the wave catastrophe solution found by Chandrasekhar in 1943 [Ballistic Research Laboratory Report No. 423 (1943)]. At $t = T/8$, our interpretation ceases to exist: at this instance, all three effectively one-dimensional shock waves emanating from each of the three sides of the initial triangle collide at the origin, and the 2D-1D correspondence between the solution of [arXiv:2011.01415] and the Damski-Chandrasekhar shock wave becomes invalid.

# 1 Introduction

## 1.1 The triangular breather phenomenon

The present work originates from the recent serendipitous experimental discovery of triangular-shaped two-dimensional (2D) breathers—periodically pulsating objects—in experiments with 2D harmonically trapped Bose condensates [1]. In this generation of experiments, it is possible to impose essentially any initial shape on the cloud. Reference [1] used uniformly filled triangles, squares, pentagons, hexagons, disks, and some other shapes as initial conditions. It was found that out of all the shapes considered, two of them—the circle and the equilateral triangle—show periodic revivals, further interpreted as *2D Gross-Pitaevskii breathers*.

    In the present work, we will concentrate on the triangular-shaped breather. A Bose condensate is initially prepared in a flat-bottomed corral in the shape of an equilateral triangle. When the condensate is subsequently released in a 2D harmonic trap, the outer edge of the condensate first starts expanding. At the same time, the flat-density patch in the center of the atomic cloud starts shrinking in area and increasing in density, while the *transition region* between the flat patch and the zero-density edge expands in size. See Fig. 1 for an illustration of the cloud geometry. It has been seen both experimentally and in Gross-Pitaevskii simulation that at a time

$$t = \frac{T}{8} \ ,$$

the central flat-density patch *disappears* and the condensate acquires hexagonal symmetry. Here and below, $T \equiv 2\pi/\omega$ is the period of the applied harmonic trap, which has frequency $\omega$.

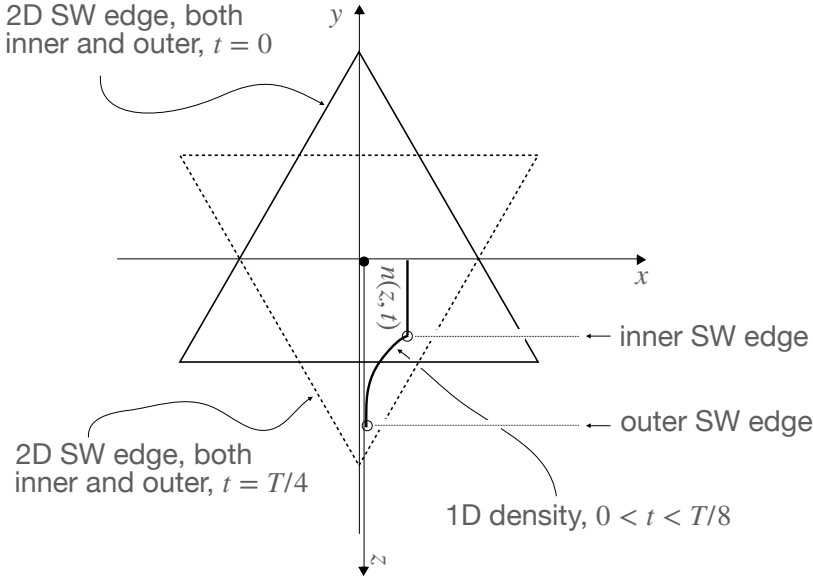

Figure 1: The geometry of the problem. We show both the initial corral (solid line) and its upside-down revival at $t = \frac{T}{4}$ (dashed line), as well as the density along the down-vertical ray ($x = 0$, $y \equiv -z < 0$), as obtained from the proposed one-dimensional theory. The one-dimensional theory describes a free-propagating Damski-Chandrasekhar shock wave (SW), controllably distorted by the harmonic confinement. At $t = T/8$, the bulk portion of the density distribution disappears and the one-dimensional theory stops being valid.

The flat patch then reappears, and at

$$t = \frac{T}{4} \;,$$

one again finds a flat-density triangular shape, but this time oriented *upside down.* At

$$t = \frac{3T}{8} \;,$$

the central flat-density patch *disappears again* and a hexagon reappears. At

$$t = \frac{T}{2} \;,$$

the condensate *returns to its initial shape.* See Fig. 1 for the general layout.

To suppress oscillations of the moment of inertia in the experiments [2], the size of the triangle was chosen in such a way that the initial trapping energy was exactly equal to the sum of the kinetic and interaction energies. In particular, this condition guarantees that the size of the upside-down triangle at $t = T/4$ is the same as the size of the original one. In fact, under the above condition, the time evolution between $t = T/4$ and $t = T/2$ is an upside-down version of the evolution between $t = 0$ and $t = T/4$.

These results are fully consistent with a numerical simulation using the Gross-Pitaevskii

equation [3]

$$i\hbar\frac{\partial}{\partial t}\psi = -\frac{\hbar^2}{2m}\Delta\psi + g|\psi|^2\psi + \frac{m\omega^2}{2}r^2\psi$$

$$\int |\psi|^2\, d^2\boldsymbol{r} = N$$

$$\psi = \psi(\boldsymbol{r},\, t)$$

(1)

Here and below, $m$ is the atomic mass, $g > 0$ is the Gross-Pitaevskii coupling constant, $N$ is the number of atoms, and $\omega$ is the trapping frequency. The initial state is

$$\psi(\boldsymbol{r},\, t = 0) = \text{ the ground state of an equilateral-triangle-shaped corral,}$$

(see [1, 4, 5]).

## 1.2   Thomas-Fermi hydrodynamics: the Shi-Gao-Zhai solution

For a slow spatial variation of the wavefunction, one may neglect the second spatial derivative of the magnitude of the wavefunction $|\psi|$ in the Gross-Pitaevskii equation (1) and arrive at the time-dependent Thomas-Fermi hydrodynamics [6]:

$$\frac{\partial}{\partial t}n + \boldsymbol{\nabla}\cdot(n\boldsymbol{v}) = 0 \qquad\qquad \text{continuity equation}$$

$$\frac{\partial}{\partial t}\boldsymbol{v} + (\boldsymbol{v}\cdot\boldsymbol{\nabla})\boldsymbol{v} = -\frac{1}{m}\boldsymbol{\nabla}(gn) - \omega^2\boldsymbol{r} \qquad \text{Euler equation,}$$

(2)

where $n(\boldsymbol{r},t) = |\psi(\boldsymbol{r},t)|^2$ is the time-dependent density profile and

$$\boldsymbol{v}(\boldsymbol{r},t) = \frac{1}{2imn(\boldsymbol{r},t)}\Big(\psi^*(\boldsymbol{r},t)\big[\boldsymbol{\nabla}\psi(\boldsymbol{r},t)\big] - \big[\boldsymbol{\nabla}\psi^*(\boldsymbol{r},t)\big]\psi(\boldsymbol{r},t)\Big)$$

is the velocity field. The initial conditions are

$$n(\boldsymbol{r},\, t = 0) = \begin{cases} n_0 & \text{for } \boldsymbol{r} \in \begin{array}{l}\text{an equilateral triangle with} \\ \text{side length } L_0 = 2\sqrt{3}R_\mu\end{array} \\ 0 & \text{otherwise}, \end{cases}$$

$$\boldsymbol{v}(\boldsymbol{r},\, t = 0) = \boldsymbol{0}\,.$$

(3)

Here, the characteristic length scale $R_\mu$ is

$$R_\mu \equiv \frac{V_\mu}{\omega}\,,$$

where

$$V_\mu \equiv \sqrt{\frac{gn_0}{m}}$$

is the characteristic velocity related to the interaction strength of the atoms and to the density. The initial triangle side, $L_0$, is chosen so that the oscillations of the moment of inertia are suppressed [2].

Thanks to an ingenious insight, the authors of Ref. [5] found a very innovative solution of the 2D Thomas-Fermi hydrodynamic equations that reproduces the "triangular breather"

phenomenon. The authors have shown that for triangular shapes, and *only* for triangular ones, there is an exact map between the ideal 2D gas with a flat phase-space density distribution (a zero-temperature "classical" Fermi gas) and the 2D Thomas-Fermi hydrodynamics. The map is valid during the time interval $0 < t < T/4$.

The solution of [5] reads

$$
\begin{aligned}
n(\boldsymbol{r},\, t) = \frac{1}{3\sqrt{3}} \frac{m}{g} \operatorname{Area} \Big[ & \\
\operatorname{Triangle} \Big[ \mathcal{V}_r(t)\, \boldsymbol{r}_{\text{down}} + \boldsymbol{v}_{0,r}(\boldsymbol{r},\, t),\ \mathcal{V}_r(t)\, \boldsymbol{r}_{\text{right}} + \boldsymbol{v}_{0,r}(\boldsymbol{r}\, t),\ \mathcal{V}_r(t)\, \boldsymbol{r}_{\text{left}} + \boldsymbol{v}_{0,r}(\boldsymbol{r},\, t) \Big] & \\
\cap & \\
\operatorname{Triangle} \Big[ \mathcal{V}_v(t)\, \boldsymbol{r}_{\text{down}} + \boldsymbol{v}_{0,v}(\boldsymbol{r},\, t),\ \mathcal{V}_v(t)\, \boldsymbol{r}_{\text{right}} + \boldsymbol{v}_{0,v}(\boldsymbol{r},\, t),\ \mathcal{V}_v(t)\, \boldsymbol{r}_{\text{left}} + \boldsymbol{v}_{0,v}(\boldsymbol{r},\, t) \Big] & \\
\Big] \, . &
\end{aligned}
\tag{4}
$$

Here Triangle[$\boldsymbol{a}$, $\boldsymbol{b}$, $\boldsymbol{c}$] is a triangle with vertices $\boldsymbol{a}$, $\boldsymbol{b}$, $\boldsymbol{c}$ and Area[$F$] is the spatial area of a geometric shape $F$; further,

$$
\begin{aligned}
\boldsymbol{r}_{\text{down}} &= \Big( 0, -\frac{1}{\sqrt{3}} \Big),\ \boldsymbol{r}_{\text{right}} = \Big( +\frac{1}{2}, +\frac{1}{2\sqrt{3}} \Big),\ \boldsymbol{r}_{\text{left}} = \Big( -\frac{1}{2}, +\frac{1}{2\sqrt{3}} \Big), \\
\mathcal{V}_r(t) &= \frac{2\sqrt{3}V_\mu}{\cot(\omega t)}, \\
\mathcal{V}_v(t) &= \frac{2\sqrt{3}V_\mu}{\tan(\omega t)}, \\
\boldsymbol{v}_{0,r}(\boldsymbol{r},\, t) &= +\frac{\boldsymbol{r}\omega}{\sin(\omega t)}, \\
\boldsymbol{v}_{0,v}(\boldsymbol{r},\, t) &= -\frac{\boldsymbol{r}\omega}{\cos(\omega t)} \, .
\end{aligned}
$$

The solution (4) of Ref. [5] perfectly reproduces the time-evolution observed in the experiment [1]. At $t = 0$ and $t = T/4$, the solution has, at the edge of the cloud, a discontinuous jump in the density. Formally, at such instances, the hydrodynamical equations (2) become invalid, being unable to properly interpret the infinite interaction force $-\boldsymbol{\nabla}(gn)$ appearing on the right-hand side of the Euler equation. Our goal is to interpret this discontinuity.

## 1.3 A side remark: the bulk density

An almost trivial observation that we will nonetheless use below is as follows. All the way to $t = T/8$, the solution (4) features a central region of a flat density $n_{\text{bulk}}(t)$. In this region, the interaction force vanishes, leaving only the force $-m\omega^2\boldsymbol{r}$ of the external trap. The atoms there are freely falling towards the center. It is easy to show that the resulting density behaves as

$$
n_{\text{bulk}}(t) = \frac{n_0}{\cos^2(\omega t)} \, .
\tag{5}
$$

Accordingly, the velocity field becomes

$$
\boldsymbol{v}_{\text{bulk}}(\boldsymbol{r},\, t) = -\omega \boldsymbol{r} \tan(\omega t) \, .
\tag{6}
$$

## 2  Damski-Chandrasekhar shock waves

The article [7] poses the following question: what are the initial conditions for a general one-dimensional (1D) set of hydrodynamic equations such that the resulting solutions can be mapped to the solutions of the inviscid Burgers' equation (i.e. of the nonlinear transport equation $\partial_t u + u \, \partial_z u = 0$) and as such show a wave catastrophe?

The one-dimensional Thomas-Fermi hydrodynamics in the *absence of a trap* reads

$$
\begin{aligned}
\frac{\partial}{\partial t} n_{1D} + \frac{\partial}{\partial z}(n_{1D} v_{1D}) &= 0 && \text{continuity equation} \\
\frac{\partial}{\partial t} v_{1D} + \left( v_{1D} \frac{\partial}{\partial z} \right) v_{1D} &= -\frac{1}{m} \frac{\partial}{\partial z}(g_{1D} n_{1D}) && \text{Euler equation}
\end{aligned}
\tag{7}
$$

Inspired by [8], the author of [7] finds that the ansatz

$$
\begin{aligned}
n_{1D} &= \frac{1}{4} \frac{m}{g_{1D}} v_{1D}^2 \\
v_{1D} &= \frac{2}{3} u
\end{aligned}
$$

turns both the continuity equation and the Euler equation into the inviscid Burgers' equation:

$$
\frac{\partial}{\partial t} u + u \frac{\partial}{\partial z} u = 0 \ .
\tag{8}
$$

Recall that if there exist any two points in space such that $z_2 > z_1$ but $u(z_2, 0) < u(z_1, 0)$, the corresponding solution $u(x, t)$ of the equation (8) is bound to undergo a wave-breaking catastrophe at a time $t^* > 0$ and ceases to exist for $t > t^*$.

The full family of solutions of (8) is given by the implicit algebraic equation

$$
u = f(z - ut) \ ,
\tag{9}
$$

with $f(\cdot)$ an arbitrary function. The only known explicit solution of the inviscid Burgers' equation (8) was given by Chandrasekhar in 1943 [9]:

$$
u_{\text{Chandrasekhar}}(z, t) = \frac{z}{t} \ ,
\tag{10}
$$

up to arbitrary temporal and spatial shifts. To interpret the solution (10) in terms of the general solution (9), consider $f(\xi) = \frac{\xi}{\delta t}$. The equation (9) would give $u(z, t) = z/(t + \delta t)$. The solution (10) will then correspond to the following limit: $u_{\text{Chandrasekhar}}(z, t) = \lim_{\delta t \to 0} z/(t + \delta t)$.

It turns out that the solution (10) gives rise to the exact solution (4), taken along a particular ray, within a limited time interval.

This Damski-Chandrasekhar shock wave, modified for our purposes, reads

$$
\begin{aligned}
n_{1D}(z, t) &= \left\{ \frac{1}{9} \frac{m}{g_{1D}} \left( \frac{z - (Z_G + V_G t)}{t} \right)^2 \right\} \times \left\{ \begin{array}{ll} 1 & \text{for} \quad z \leq Z_G + V_G t \\ 0 & \text{otherwise} \end{array} \right\} \\
v_{1D}(z, t) &= \left\{ \frac{2}{3} \left( \frac{z - (Z_G + V_G t)}{t} \right) + V_G \right\} \times \left\{ \begin{array}{ll} 1 & \text{for} \quad z \leq Z_G + V_G t \\ \text{undeterm.} & \text{otherwise} \end{array} \right\}
\end{aligned}
\tag{11}
$$

where we have introduced an arbitrary Galilean boost $V_G$ and translation $Z_G$. Note that we have set the right spatial half of the Chandrasekhar solution (10) to zero; it can be shown that such a truncated function remains a solution of the inviscid Burgers' equation (8). Indeed, such truncation leaves the field $u(z, t)$ spatially and temporally continuous; only the spatial derivative at the origin becomes discontinuous. Since the inviscid Burgers' equation (8) is of first order in the coordinate, no new terms appear on its right hand side after the discontinuity is introduced. Also, the $z > 0$ part of the truncated field, $u(z, t) = 0$ is a valid solution of (8), and hence the whole of the truncated field, $u(z, t) = (z/t)\,\theta(-z)$ is a valid solution of (8).

We observe the following:

1. (a) At the zero-density point

$$Z_{n=0}(t) = Z_G + V_G t \ ,$$

   where

$$n_{1D}(Z_{n=0}(t), t) = 0 \ ,$$

   the force $-\partial_z(g_{1D}n_{1D})$ is zero at all times $t > 0$.
   (b) The particle velocity at this point coincides with the velocity of the point itself:

$$v_{1D}(Z_{n=0}(t), t) = V_G.$$

   These two observations are consistent with the fact that the point $Z_{n=0}(t)$ moves at a constant velocity. Nonetheless, at the moment, it is not clear if either (a) or (b) is a generic property. Looking ahead, both of them are guaranteed by a map to an ideal gas [5]. Indeed, on the ideal gas end, the edge $Z_{n=0}(t)$ is represented by a single free particle. In the absence of a map, the kinematics of the $Z_{n=0}(t)$ edge has to be reevaluated.

2. The solution (11) remains exact at all times, not only in the beginning of the evolution.

3. Let us select a density value $n_{0,1D}$. The point at which the density reaches this value,

$$Z_{n=n_{0,1D}}(t) = Z_G + \left(-3\sqrt{\frac{g_{1D}n_{0,1D}}{m}} + V_G\right) t \ ,$$

   such that

$$n_{1D}(Z_{n=n_{0,1D}}(t), t) = n_{0,1D} \ ,$$

   also moves at the constant velocity

$$V_{n=n_{0,1D}}(t) = -3\sqrt{\frac{g_{1D}n_{0,1D}}{m}} + V_G \ .$$

Now, select a velocity value $v_{0,1D}$ and require that the solution (11) reaches this velocity $v_{0,1D}$ at the point $Z_{n=n_{0,1D}}(t)$. Interestingly, this can be fulfilled at all times, simply by setting

$$V_G = 2\sqrt{\frac{g_{1D}n_{0,1D}}{m}} + v_{0,1D} \ .$$

Now, the solution (11) can be amended as follows (see also Fig. 1):

$$
n_{1D}(z,\,t) = \left\{ \begin{array}{ccc} n_{0,1D} & \text{for} & z \leq Z_{\text{inner}}(t) \\ \left\{ \frac{1}{9}\frac{m}{g_{1D}}\left(\frac{z-Z_{\text{outer}}(t)}{t}\right)^{2}\right\} & \text{for} & Z_{\text{inner}}(t) \leq z \leq Z_{\text{outer}}(t) \\ 0 & \text{for} & z \geq Z_{\text{outer}}(t) \end{array} \right\},
$$

$$
v_{1D}(z,\,t) = \left\{ \begin{array}{ccc} v_{0,1D} & \text{for} & z \leq Z_{\text{inner}}(t) \\ \frac{2}{3}\left(\frac{z-Z_{\text{outer}}(t)}{t}\right)+V_{\text{outer}} & \text{for} & Z_{\text{inner}}(t) \leq z \leq Z_{\text{outer}}(t) \\ \text{undetermined} & \text{for} & z \geq Z_{\text{outer}}(t) \end{array} \right\},
$$
(12)

with the positions and the velocities of the outer and inner edges given by

$$
V_{\text{outer}} = 2\sqrt{\frac{g_{1D}n_{0,1D}}{m}} + v_{0,1D}\,,
$$
(13)

$$
Z_{\text{outer}}(t) = Z_{\text{edge},0} + V_{\text{outer}}t\,,
$$
(14)

$$
V_{\text{inner}} = -\sqrt{\frac{g_{1D}n_{0,1D}}{m}} + v_{0,1D}\,,
$$
(15)

$$
Z_{\text{inner}}(t) = Z_{\text{edge},0} + V_{\text{inner}}t\,,
$$
(16)

where $Z_{\text{edge},0}$ is the arbitrary initial position of the shock wave front, infinitely narrow at this instance.

Two more observations:

4. Velocity $v_{0,1D}$ with which the atoms move at the inner edge is different from the velocity $V_{\text{inner}}$ of the edge itself.

5. The formula (15) for the velocity of the inner edge of the shock wave front can be proven for any discontinuity in the derivative of the density, using matter conservation alone. However, this conclusion is only valid in one dimension: it can be shown that additional terms in the continuity equation destroy this relationship in the case of non-straight 2D edges.

## 3   A general map between hydrodynamic solutions, induced by scale invariance

Pitaevskii and Rosch discovered a particular symmetry of 2D Bose-condensates [2]. This symmetry stems from the fact that in two dimensions, the coupling constant $g$ in Ref. (1) has the same dimensionality as the diffusion constant $\hbar^2/(2m)$ appearing in the kinetic energy and as such does not induce a length scale. As a result, the following three observables form a closed algebra: the Hamiltonian, the moment of inertia (proportional to the hyperradius), and the generator of scaling transformations. Empirical consequences are that (a) the dynamics of the moment of inertia separates from that of the rest of the system, and (b) there emerges an additional integral of motion, namely the Casimir invariant for the above algebra. These properties allow us to relate the dynamics of any two systems that have the same hyperangular dynamics—the dynamics complementary to the dynamics of the hyperradius—but different

hyperradial one and, more generally, different dependence of their Hamiltonians on the hyperradius.

Let us first introduce the hyperradius $\mathcal{R}(t)$:

$$\mathcal{R}(t) \equiv \left( \int n(\boldsymbol{r},\, t)\, r^2\, d^2\boldsymbol{r} \right)^{\frac{1}{2}} . \tag{17}$$

The new integral of motion that is preserved by the hydrodynamic equations (2) in the 2D case is represented by the square of a generalized hyperangular momentum:

$$\mathcal{L}^2 \equiv 2m\mathcal{R}^2(t)\left\{ E_{\text{kinetic-hyperangular}}(t) + E_{\text{interaction}}(t) \right\} , \tag{18}$$

where

$$E_{\text{kinetic-hyperangular}}(t) = \int n(\boldsymbol{r},\, t)\, \frac{m\boldsymbol{v}^2(\boldsymbol{r},\, t)}{2}\, d^2\boldsymbol{r} - \frac{m\dot{\mathcal{R}}^2(t)}{2}$$

and

$$E_{\text{interaction}}(t) = \int \frac{1}{2}\, g n^2(\boldsymbol{r},\, t)\, d^2\boldsymbol{r}$$

are respectively the kinetic hyperangular and interaction energies.

The dynamics of the hyperradius is governed by the equation of motion

$$\ddot{\mathcal{R}}(t) = \frac{\mathcal{L}^2}{m^2\mathcal{R}^3(t)} - \omega^2\mathcal{R}(t) . \tag{19}$$

The motion generated by equation (19) is an isochronous (meaning that the period does not depend on the energy) but polychromatic oscillation of universal base frequency $2\omega$. A stationary fixed point of (19) is

$$\mathcal{R}_0 = \sqrt{\frac{\mathcal{L}}{m\omega}} . \tag{20}$$

Note that at this point, the sum of the kinetic hyperangular and interaction energies equals the trapping energy: $E_{\text{kinetic-hyperangular}} + E_{\text{interaction}} = E_{\text{trapping}}$, where

$$E_{\text{trapping}}(t) = \int n(\boldsymbol{r},\, t)\frac{m\omega^2 r^2}{2}\, d^2\boldsymbol{r} .$$

At the level of the hydrodynamic equations (2), the map between two motions sharing the same hyperangular dynamics looks as follows. Consider two sets of the 2D hydrodynamic equations (2), generally corresponding to two different trapping frequencies, $\omega_1$ and $\omega_2$ but with the same coupling constant $g$:

$$\begin{aligned} \frac{\partial}{\partial t_{1,2}}n_{1,2} + \boldsymbol{\nabla}_{1,2}\cdot(n_{1,2}\,\boldsymbol{v}_{1,2}) &= \boldsymbol{0} \\ \frac{\partial}{\partial t_{1,2}}\boldsymbol{v}_{1,2} + (\boldsymbol{v}_{1,2}\cdot\boldsymbol{\nabla}_{1,2})\,\boldsymbol{v}_{1,2} &= -\frac{1}{m}\boldsymbol{\nabla}_{1,2}(g\,n_{1,2}) - \omega_{1,2}^2 r_{1,2} \end{aligned} \tag{21}$$

where $\boldsymbol{\nabla}_{1,2} \equiv \partial/\partial\boldsymbol{r}_{1,2}$. It can be straightforwardly verified that there is a one-to-one correspondence between the solutions of the first and the second sets, given by

$$\mathcal{R}_1^2(t_1)\, n_1(\boldsymbol{r}_1,\, t_1) = \mathcal{R}_2^2(t_2)\, n_2(\boldsymbol{r}_2,\, t_2)$$

$$\mathcal{R}_1(t_1)\left(\boldsymbol{v}_1(\boldsymbol{r}_1,\, t_1) - \boldsymbol{r}_1\frac{d}{dt_1}\ln[\mathcal{R}_1(t_1)]\right) = \mathcal{R}_2(t_2)\left(\boldsymbol{v}_2(\boldsymbol{r}_2,\, t_2) - \boldsymbol{r}_2\frac{d}{dt_2}\ln[\mathcal{R}_2(t_2)]\right), \tag{22}$$

where

$$\frac{\boldsymbol{r}_1}{\mathcal{R}_1(t_1)} = \frac{\boldsymbol{r}_2}{\mathcal{R}_2(t_2)}$$

$$\frac{dt_1}{\mathcal{R}_1^2(t_1)} = \frac{dt_2}{\mathcal{R}_2^2(t_2)}, \tag{23}$$

with

$$t_1 = 0 \Leftrightarrow t_2 = 0 . \tag{24}$$

## 4  A particular scale-invariance-induced map to be used

A particular case of the general map (22)–(24) is given by

$$\begin{aligned} &\omega_1 = \omega &&\omega_2 = 0 \\ &n_1(\boldsymbol{r}_1,\, 0) = n_{\text{in}}(\boldsymbol{r}_1) &&n_2(\boldsymbol{r}_2,\, 0) = n_{\text{in}}(\boldsymbol{r}_2) \\ &\boldsymbol{v}_1(\boldsymbol{r}_1,\, 0) = \boldsymbol{0} &&\boldsymbol{v}_2(\boldsymbol{r}_2,\, 0) = \boldsymbol{0} , \end{aligned} \tag{25}$$

where $n_{\text{in}}(\boldsymbol{r})$ is the initial density, the same for both systems. Notice that the two systems share the same initial value of the hyperradius,

$$\mathcal{R}_1(0) = \mathcal{R}_2(0) \equiv \mathcal{R}_{(0)} ,$$

and the same value of the Casimir invariant,

$$\mathcal{L}_1 = \mathcal{L}_2 \equiv \mathcal{L}_{(0)} .$$

We will identify the System 1 with the system described by Eqs. (2), subject to the initial conditions (3). Recall that these initial conditions were chosen in such a way that the hyperradius $\mathcal{R}_1$ resides at the stationary point:

$$\mathcal{R}_1(t_1) = \mathcal{R}_{(0)} \equiv \sqrt{\frac{\mathcal{L}_{(0)}}{m\omega}} .$$

The identification of System 1 is completed by setting

$$(n,\, \boldsymbol{v}) = (n_1,\, \boldsymbol{v}_1)$$
$$(\boldsymbol{r},\, t) = (\boldsymbol{r}_1,\, t_1) .$$

As for System 2, we will take it to be the same as System 1 except that there is no trapping potential. We get

$$
\begin{aligned}
\frac{\mathcal{R}_2}{\mathcal{R}_{(0)}} &= \sqrt{1 + (\omega t_2)^2} = \frac{1}{\cos(\omega t)} \\
\boldsymbol{r}_2 &= \sqrt{1 + (\omega t_2)^2}\, \boldsymbol{r} = \frac{\boldsymbol{r}}{\cos(\omega t)} \\
t_2 &= \frac{1}{\omega}\tan(\omega t)
\end{aligned}
\qquad , \tag{26}
$$

and, accordingly,

$$
\begin{aligned}
n(\boldsymbol{r},\, t) &= \frac{1}{\cos^2(\omega t)} n_2(\boldsymbol{r}_2(\boldsymbol{r},\, t),\, t_2(t)) \\
\boldsymbol{v}(\boldsymbol{r},\, t) &= \frac{1}{\cos(\omega t)}\left(\boldsymbol{v}_2(\boldsymbol{r}_2(\boldsymbol{r},\, t),\, t_2(t)) - \omega\boldsymbol{r}_2(\boldsymbol{r},\, t)\sin(\omega t)\cos(\omega t)\right)
\end{aligned}
\qquad . \tag{27}
$$

## 5 The Shi-Gao-Zhai solution *vs.* Damski-Chandrasekhar shock waves

Let us emphasize that System 2, subject to the map (25)–(27), describes *free propagation* from the initial condition (3), depicted in Fig. 1 as a solid line. We now focus our attention to the center of the base of the initial triangle, at $(x = 0,\, y = -L_0/(2\sqrt{3}) = -R_\mu)$. In free propagation from a triangle, the left and right vertices bounding the base cannot have an immediate effect on the dynamics in the center. As a result, for a period of time, the propagation in the base center, under the "free" System 2, will effectively be a free one-dimensional propagation. This is the point where the Damski-Chandrasekhar shock wave emerges as a description of the dynamics.

Let us make the following association:

$$
\begin{aligned}
\frac{g n_2((x_2 = 0,\, y_2 = -z_2),\, t_2)}{m} &= \frac{g_{1D} n_{1D}(z_2,\, t_2)}{m} \\
(\boldsymbol{v}_2((x_2 = 0,\, y_2 = -z_2),\, t_2))_y &= -v_{1D}(z_2,\, t_2)
\end{aligned}
\qquad ,
$$

where $n_{1D}(z,\, t)$ and $v_{1D}(z,\, t)$ describe the Damski-Chandrasekhar shock wave (12), with

$$
\begin{aligned}
Z_{\text{edge},0} &= L_0/(2\sqrt{3}) = R_\mu \equiv \frac{V_\mu}{\omega} \\
\frac{g_{1D} n_{0,1D}}{m} &= \frac{g n_0}{m} \equiv V_\mu^2 \\
v_{0,1D} &= 0
\end{aligned}
\qquad . \tag{28}
$$

The bulk density $n_{0,1D}$ remains constant both in space and in time. The front of the shock wave is a half-parabola, with a center at

$$
Z_{\text{outer}}(t_2) = R_\mu(1 + 2\omega t_2)
$$

and the bulk interface at

$$
Z_{\text{inner}}(t_2) = R_\mu(1 - \omega t_2)\ .
$$

At $t_2 = 1/\omega$, the inner edge of the shock wave front reaches the origin and the one-dimensional theory collapses. Now observe that according to the map (26), $t_2 = 1/\omega$ corresponds to the actual time of $t = T/8$, which is exactly the instance when the bulk disappears in the exact 2D solution (4). In general, the shock wave (12)–(16), with the association (28), under the map (25)–(27), can be shown to reproduce the solution (4) at $(x = 0, y = -z)$ exactly, for a period of time $0 \leq t \leq T/8$. Figure 2 corroborates this correspondence.

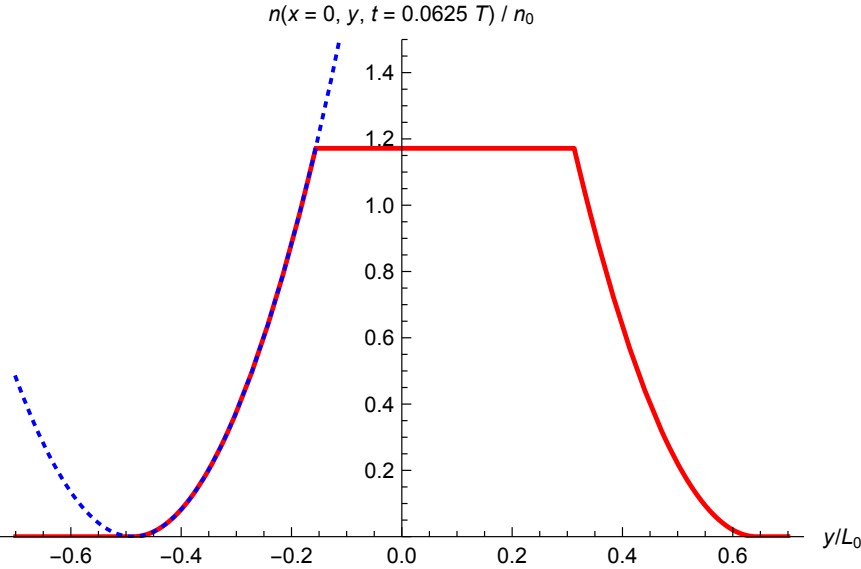

Figure 2: The shock wave theory (blue, dashed) vs exact hydrodynamics (red). The density is plotted at $t = T/16$, along the vertical symmetry axis of the triangle.

We expect that the other points on the base of the original triangle, along with their counterparts on the other two sides, will also behave as one-dimensional shock waves, but with a solution that stops being valid before $T/8$.

# 6    Discussion and summary

We have interpreted the exact solution [5] for the triangular breather observed in the experiment [1] in terms of the Gross-Pitaevskii shock waves introduced by Damski [7]. More specifically, under the transformation discovered in [7], the $t = 0$ singularity of the original problem becomes, *verbatim*, the initial condition for the wave-breaking catastrophe solution of the inviscid Burgers' equation (also commonly referred to as the nonlinear transport equation) found by Chandrasekhar in 1943 [9]. This interpretation remains valid and exact for all times in the $0 < t < T/8$ interval.

Chandrasekhar's catastrophe at $t = 0$ is consistent with a loss of the Fermi-Bose connection observed in [5]. It is also consistent with the fact that at $t = 0$, Chandrasekhar's solution breaks the time-reversal invariance that is dictated by the underlying Gross-Pitaevskii equation [1,4,5]. Namely, at $t = 0$, the Galilean boost $V_G$ (which determines the velocity of the outer edge of the shock wave (13-14); see (11)) *reverses sign* and thus undergoes a sudden jump. Such a discontinuity can not be supported by the hydrodynamic equations, signifying a failure thereof.

A related phenomenon occurs at the time $t = T/8$. This is the moment when the inner edge (15-16) of the shock wave reaches the origin, where it meets the two other shock wave edges, originating from the two other sides of the initial triangle. At this instant, the region occupied by the bulk (5-6) shrinks to a point. Again, the time-reversal invariance suggested by both the experiment and the Gross-Pitaevskii numerics implies that at $t = T/8$, the bulk velocity gradient in (6) *reverses sign*, along with the velocity of the inner edge (15), thus signifying another breakdown of the hydrodynamic description.

Curiously, at $t = T/8$, the Fermi-Bose map [5] remains valid. It is only at $T/4$, that the map starts producing results different from the Gross-Pitaevskii predictions and requires an abrupt parameter update.

# Acknowledgements

We are immeasurably grateful to Jean Dalibard for numerous discussions and as well for providing access to unpublished numerical data. We also thank Zhe-Yu Shi and Bogdan Damski for useful discussions.

**Funding information**   This work was supported by the NSF (Grants No. PHY-1912542, and No. PHY-1607221) and the Binational (U.S.-Israel) Science Foundation (Grant No. 2015616). J.T.'s project PID2019-106290-C22 is financed by Ministerio de Ciencia e Innovación de España. M.G. is partially supported by the Spanish grant PGC2018-098676-B-I00 (AEI/FEDER/UE) and the Juan de la Cierva-Incorporación fellowship IJCI-2016-29071. G.E.A. acknowledges financial support from the Spanish MINECO (FIS2017-84114-C2-1-P), and from the Secretaria d'Universitats i Recerca del Departament d'Empresa i Coneixement de la Generalitat de Catalunya within the ERDF Operational Program of Catalunya (project QuantumCat, Ref. 001-P-001644).

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

—