# Peer review of "Triangular Gross-Pitaevskii breathers and Damski-Chandrasekhar shock waves"

_SciPost Physics_

## Round 5 · Referee Report · Anonymous (Referee 1) · 2021-4-19

Strengths

1- There is a clever idea behind whole this work supported by physically- and technically-significant results on a hot research topic.

2- The presentation of these findings is concise, without unnecessary discussions and propaganda. The paper is easy to follow.

3- The plots are adequate, especially the first one, cleverly putting different pieces of the considered setup into one place.

4- The reference to Chandrasekhar’s paper is real nice, I have never seen it cited before in the context of cold atom shock waves.

Weaknesses

1- The manuscript does not discuss the role of the quantum pressure term in the dynamics of the studied system.

Report

The manuscript “Triangular Gross-Pitaevskii breathers and Damski-Chandrasekhar shock waves” analyzes the outcome of the recent, already well-known, experiment of the Paris group from the perspective of the shock wave phenomenon.

It was found in this fascinating experiment that initially triangular- and disc-shaped spatially- flat two-dimensional clouds of atoms undergo robust periodic dynamics in a harmonic potential. This work triggers (at the very least) the following two outstanding questions

$\bullet$ why other initial shapes evolve differently,

$\bullet$ how dynamics of two-dimensional condensates, starting from a flat density profile bounded by a sharp edge, proceed.

The first of these questions is interestingly analyzed in the very recent, possibly seminal, paper of Shi, Gao & Zhai. The second one is discussed in the manuscript that is under review here.

In this manuscript, the authors very interestingly argue that some aspects of the dynamics of the above-mentioned condensate can be successfully explained by invoking the shock wave phenomenon. They have written down an elegant analytical 1+1 dimensional shock wave solution, described the mapping/conditions allowing for its application to the 2+1 dimensional flat triangular cloud of atoms, and explained where and when it exactly reproduces the elegant result of Shi, Gao & Zhai.

This way, in my opinion, they have added crucial new insights to our understanding of both the outcome of the Paris experiment and the content of the Shi, Gao & Zhai result. Indeed, neither the paper of the Paris group nor the one of Shi, Gao & Zhai (the ones cited in the manuscript) invokes the shock wave phenomenon as far as I can see. This makes the manuscript of Olshanii et al. physically-interesting. It can be also argued that this work is technically- interesting as well, due to the above-mentioned solution that is worked out there. It is my opinion that these two aspects make the manuscript under review well-worth publication in a respectable journal such as SciPost.

As far as technical remarks go, I would like to mention that I have been able to check some equations. In particular, I have quickly verified all equations from Sec. 4, finding them to be correct. A few misprints, which I have been able to locate, are listed along with some remarks/suggestions in the "Requested changes" section.

Summarizing, it is my opinion that after small corrections, the manuscript should be published in SciPost. It does present original insights/results on a hot research topic and it is well- structured. Moreover, it may be worth to mention that it seems to be reasonable to assume that this work will trigger interesting subsequent studies. One of the reasons for saying so is that the question of why the discussed hydrodynamic solution seems to work so well, despite the fact that it is written without taking into account the quantum pressure term, remains to be resolved.

Requested changes

1- Given the fact that whole Sec. 1 is composed of subsection 1.1, one may consider removal of the subsection title.

2- One may consider providing explicit definition of SW and SGZ (despite the fact that these terms should be understandable per se).

3- One may indicate below eq. (1) that g is greater than zero, which is tacitly assumed in the manuscript.

4- The last equation of Sec. 1 mentions “ground state of an equilateral-triangle-shaped corral” while the text above it says that we deal with “a uniformly filled equilateral triangle”. The two statements are not equivalent due to the boundary effects “supported” by eq. (1).

5- Bold zero in the continuity eq. (2) should be replaced by “normal” zero.

6- “Normal” $\nabla$ in the equation for the velocity field, the one between eqs. (2) and (3), should be replaced by bold $\nabla$.

7- Odd-looking “and” in-between equations on p. 5 and in eq. (14) perhaps could be removed.

8- “Euler equation” appears by the end of Sec. 2, but it is defined in eq. (7) of Sec. 4. Similarly, “force” is discussed on p. 5, but a precise meaning to it is given on p. 7. One may consider explicit introduction of these terms around eq. (2).

9- It seems to me that the factor of “$\omega$” may be lost in eq. (6), the dimension of the expression on its right-hand side is wrong.

10- It might be worth to mention that both the continuity equation and the Euler equation, the ones listed as (7), reduce to eq. (8) under the discussed mapping.

11- I am not sure what the sentence “Formally, this solution is the $\delta t\to0$ limit of the solution generated by ...” exactly means. Could this be clarified?

12- It looks like “also at constant velocity”, which can be found on p. 7, should be replaced by something like “also moves at a constant velocity”.

13- Time argument might be added to $\dot{\cal R}$ in the first equation on p. 9.

14- On the right-hand side of the equation on p. 10, the one right above the sentence “The identification of System 1 is completed by setting”, a square root seems to be missing.

15- It seems to me that in eq. (27), the one where the right-hand side is just $V_\mu$, one should have $V_\mu^2$ instead.

  • validity: high
  • significance: high
  • originality: top
  • clarity: high
  • formatting: good
  • grammar: excellent

Author:  Vanja Dunjko  on 2021-05-05  [id 1408]

(in reply to Report 1 on 2021-04-19)

We thank the Referee for the positive review and encouragement. We have implemented all the recommendations the Referee has made, and we thank the Referee for those as well.

---

## Round 5 · Referee Report · Anonymous (Referee 2) · 2021-4-23

Strengths

1 - Nice problem, clearly exposed.

2 - An elegant combination of interesting and nontrivial concepts applied to a beautiful experiment.

Weaknesses

none

Report

This is a theoretical work based on a recent beautiful experiment of Dalibard's group in Paris (ref.[1]) and on a previous theoretical article by Shi, Gao and Zhai (ref.[5]). The experiment proved the existence of triangular-shaped breathers in 2D harmonically trapped Bose condensates, while the authors of ref.[5] found a quite surprising map between the hydrodynamic equations governing these 2D breathers and those describing a high-density zero-temperature triangular-shaped cloud of fermions. Such a mapping, however, has a problem at the initial time, since it implies an infinite force at the cloud edge. In the present work, a solution is found to this problem involving shock waves and making a connection with an old prediction of Chandrasekhar. The three papers together provide a clean example of clever application of general concepts of symmetry, scale invariance and nonlinearity in the collective dynamics of many-body quantum systems, accessible to experimental investigation.
The quality of the paper is high, both for the relevance of the results and the clarity of presentation.
I recommend publication after very minor changes (see below).

Requested changes

1 - in the definition of the velocity v, I suggest to enclose the gradient of Psi and Psi* in parenthesis, in order to avoid misunderstanding, especially in the second term.

2 - near the end of section 4, replace "clout" with "cloud"

3 - after eq.(9), replace "Reacall" with "Recall"

4 - at the beginning of section 5, replace "Pitaevski" with "Pitaevskii"

5 - six lines after, correct "separates form..."

6 - after eq.(10) the authors say: "it can be shown that such a truncated function remains a solution of the inviscid Burgers’ equation". Is it easy to demonstrate? Should the reader take it for granted?

  • validity: top
  • significance: high
  • originality: high
  • clarity: top
  • formatting: excellent
  • grammar: excellent

Author:  Vanja Dunjko  on 2021-05-05  [id 1409]

(in reply to Report 2 on 2021-04-23)

We thank the Referee for the positive review and encouragement. We have implemented all the recommendations the Referee has made, and we thank the Referee for those as well.

---

## Editorial Decision

resubmitted